# One-Dimensional Mathematical Model for Kayak Propulsion

**Diego Delgado** [1,*] and **Camilo Ruiz** [2]

1   Departamento de Matemáticas, Universidad de Oviedo, 33003 Oviedo, Spain
2   EMC3 Research Group, Department of Maths and Science Education, Universidad de Salamanca, 37008 Salamanca, Spain; camilo@usal.es
*   Correspondence: ddb78s@gmail.com or delgadodiego@uniovi.es

**Abstract:** The displacement of a sprint kayak can be described by a one-dimensional mathematical model, which, in its simplest case, is analogous to the free-fall problem with quadratic drag and constant propulsion. To describe realistic cases, it is necessary to introduce a propulsion capable of reproducing the characteristics of the kayak stroke, including periodicity, average force and effects of stroke frequency, among others. Addressing the problem in terms of a Fourier series allows us to separate the equation into two parts, one of which is equivalent to the constant propulsion case and results in an asymptotic expression, while the second accounts for the periodic contributions. This approach allows us to solve several cases of interest: to propose a quadrature rule for the asymptotic part that allows fast estimations; to compare results with the literature; and finally to propose a general mathematical method for this problem which could help to understand some key strategies in the kayak race.

**Keywords:** kayak propulsion; analytic model; mathematical model; Fourier series; periodic propulsion; reduced dimensionality models; applied physics; differential equations





## 1. Introduction

Professional sports have seen important transformations due to the use of new data collection techniques to create models and optimize performance. These data-driven models provide important insight into the sports but they may be hard to extend to different situations. Analytic models provide an alternative approach, which allows to study the sports from basic principles and provides insights on the main drivers of performance and strategies to improve it. Their results are expressed in terms of simple physical constants and can be generalized to different situations.

In the case of competition canoeing, many different aspects have been studied in recent years, which reflects the complexity of this sport. The hydrodynamic aspects of the propulsion have been investigated extensively; the hydrodynamic drag or its components was analyzed both computationally and experimentally [1–6], even under uniform displacement conditions. The paddle [7] or stroke [8] hydrodynamics were also analyzed; correlations between the stroke frequency and hull speed [9–11], the stroke force versus time functional relation [9,12] or the hull–stroke interaction [13,14] were found. There are also studies on the forces applied to the paddle blade and the power, accelerations and velocities that it produce in the hulls [9,12,15,16]. Additionally there are kinematics kayak models: the one of six degree of freedom proposed by Leroyer [17] or the one introduced by Begon [18] to calibrate a kayak ergometer in the most precise way.

These papers show the importance of understanding correctly hydrodynamic drag coefficients, the effect of the forces on the paddler–hull system, and the functional relationships of the stroke on the kayak to model the races and provide optimization routes for the materials of the equipment, the techniques or the strategies to increase performance.

In the present work, we propose a solvable one-dimensional mathematical model for kayak displacement as a function of the applied force. Our model allows to connect the



physics of the system to quantities relevant to the race and the performance of the athlete. It also provides a tool to analyze and develop strategies in different race scenarios.

The model is characterized by periodic propulsion and constant hydrodynamic drag and mass, which are the only relevant parameters of the kayak in our model, and are useful to analyze different aspects of the race. We introduce different functional relations in order to model the stroke and with it, the propulsion. The analytical and numerical resolution of such a model will allow us to find the good parameters to characterize the kayak advance, and to relate the analytical model with previous experimental studies. Another consequence is the description of a general mathematical technique to solve both analytically or numerically the problem of a boat propelled by a periodic force. The results here presented could have implications not only in this sport, but also in others characterized by periodic propulsion and time-independent drag.

The value of this new analytical model that we introduce is that: (i) it is built from first principles, and the reduced dimensionality allows us to model the main drivers of the system. (ii) The analytical model allows us to write quadrature relations that are useful to describe the race in simple terms, using relevant physical constants. (iii) The analytical expressions are general and allow us to use and compare different paddle models to analyze the race and understand the relevant drivers of performance; and finally, (iv) it can be used to analyze real data from races and relate the performance to simple physical constants that are relevant and easy to understand.

All these characteristics are in contrast to numerical models that can be used to integrate numerically the differential equation, which can be more accurate but cannot be generalized, and do not provide simple physical pictures to understand and improve the performance.

## 2. Mathematical Model

The fluid–hull–paddler system propelled by periodic strokes is a complex mechanical problem. A complete formulation of the problem would involve a coupled fluid dynamics model for the medium together with a dynamic model for the hull–paddler system. Some models exists for this system, using different approaches: that of Leroyer et al. [17] by using computational fluid dynamics (CFD), or that of Harrison et al. [13] by means of smoothed particle hydrodynamics (SPH). In essence, these implementations seek to computationally reproduce the real scenario as faithfully as possible, which makes use of large computational resources. Complementary to these expensive computational models are the reduced dimensionality models, which describe the substantial elements of the problem and can have analytical solutions, which allows to obtain general dynamics in simple mathematical terms, providing new perspectives of the problem. This is the approach followed in this work.

### 2.1. Main Equation

The complex fluid–hull–paddler system can be simplified using a dynamic model for kayak displacement. The main simplifications in our model are to replace the fluid by a general hydrodynamic drag term, the propulsion by a time-dependent force and to bound the displacement to one dimension—that of the kayak's forward movement. These simplifications preserve the main dynamics of the kayak motion and allow us to understand the most important mechanisms at play. The remaining displacements and rotations of the kayak are considered secondary, and the equation governing the dynamics is Newton's second law, written in the following form:

$$m\dot{v} + d \cdot v^2 = f \tag{1}$$

where $m$ is the mass, $\dot{v} = \frac{dv}{dt}$ is the acceleration, the terms $d \cdot v^2$ corresponds with the hydrodynamic drag, and $f = f(t)$ is the propulsion force that, in its general form, is a function of time. Similar models were proposed by other authors to describe the movement in non-linear media, especially ballistic problems in sport [19,20].

### 2.2. Constant Force Solution

The simplest case is the one where the force is constant, and the differential equation takes an analogous form to the free fall problem. We rewrite it in the following form:

$$m\dot{v} = f \cdot \left(1 - \frac{d}{f}v^2\right).$$ (2)

which could be integrated to reach the result:

$$v = \sqrt{\frac{f}{d}}tanh\left(\sqrt{\frac{d}{f}}\left(\frac{f}{m}t + K\right)\right) \quad \text{with} \quad K = \sqrt{\frac{f}{d}}tanh^{-1}\left(\sqrt{\frac{d}{f}}v(0)\right)$$ (3)

where the $K$ constant accounts for the initial conditions. This is a known result, for example, in ballistic problems applied to sport [20]. Figure 1 represents the above function starting from the rest; the horizontal asymptote of this solution represents the limiting velocity approached by the solution, with value $\sqrt{f/d}$, also known as terminal velocity [20], where the constant force and drag coefficient appear but not the mass. The mass appears in the denominator of the coefficient that accompanies the time variable in the hyperbolic tangent function; the higher the mass, the equally limiting the velocity, but the longer the transient to it and vice versa. Figure 2 represents this situation and uses three representative masses of the problem and realistic coefficients.

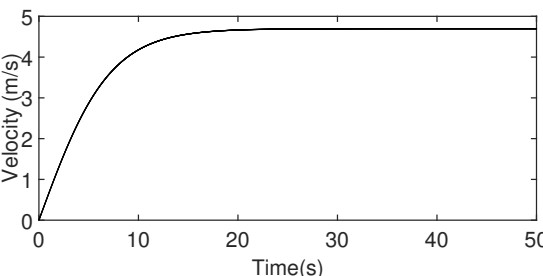

**Figure 1.** Kayak velocity versus time, $m$ = 90 kg, $f$ = 60 N, $d$ = 2.72 kg/m.

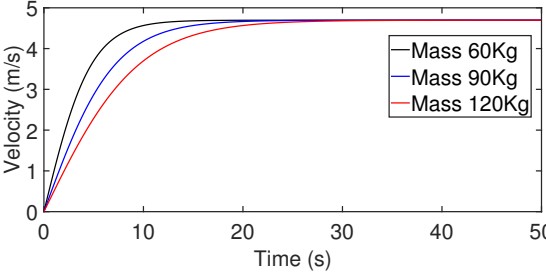

**Figure 2.** Kayak velocity versus time for three masses (see legend); $f$ = 60 N, $d$ = 2.72 kg/m.

In hydrodynamic towing tests and numerical calculations to estimate the hydrodynamic drag, it is common to assume a constant propulsive force and to adjust values of force versus velocity squared to obtain the coefficient. As the hyperbolic tangent tends to one as time tends to infinity, the limiting velocity can be written as follows:

$$v_{limit} = \lim_{t \to \infty} v(t) = \sqrt{\frac{f}{d}}$$ (4)

and clearing over this expression, we obtain the following:

$$f = d \cdot v_{limit}^2$$ (5)

which corresponds to the expression used in the hydrodynamic tests mentioned above.

### 2.3. Derivation of Relevant Quantities

The constant force, or asymptotic solution (Equation (3)), is generic for any stroke model, which we will see later, and the constant $K$ is zero if the kayak starts from rest. It can be observed that the described curve can be separated into an acceleration interval followed by an interval, where the velocity is constant. In this section, we assume that the first interval corresponds to a uniform acceleration followed by a uniform velocity interval, which in turn we will decompose into two intervals: one that we will call the stabilization interval followed by a uniform velocity interval. Likewise, we introduce a rule to approximate them in a simple way.

The area under the velocity–time curve (Figure 1) corresponds to the distance covered by the kayak. We start by finding a point on the acceleration interval such that it approximates this distance by a uniform acceleration interval; it will therefore be a quadrature rule. Let $A$ be such a point. Then, the following holds:

$$v_a = a \cdot \sqrt{\frac{f}{d}} \qquad \text{with} \qquad 0 < a < 1. \tag{6}$$

This is the velocity reached at this moment, where $a$ represents a fraction over the limiting velocity. Substituting this expression in (3), and clearing the time, we obtain the following:

$$t_a = \frac{m}{f} \sqrt{\frac{f}{d}} tanh^{-1}(a) = \frac{m}{2f} \sqrt{\frac{f}{d}} \ln \frac{1+a}{1-a} \tag{7}$$

where we have used the following identity $tanh^{-1}(x) = \frac{1}{2} \ln \left( \frac{1+x}{1-x} \right)$. The mean acceleration $< a >$ between the initial instant and $t_a$ writes as follows:

$$< a >= \frac{v_a}{t_a} = \frac{a \cdot \sqrt{\frac{f}{d}}}{\frac{m}{2f} \sqrt{\frac{f}{d}} \ln \frac{1+a}{1-a}} = \frac{2f}{m} \frac{a}{\ln \frac{1+a}{1-a}}. \tag{8}$$

The cut-off point between the constant acceleration interval and the constant velocity displacement gives us the transition point between both intervals, and is obtained by equating both straight lines, that is, as follows:

$$v_{limit} =< a > \cdot t_{accel} \qquad \rightarrow \qquad t_{accel} = \frac{v_{limit}}{< a >} = \frac{\ln \left( \frac{1+a}{1-a} \right) \cdot m}{a \cdot 2 \cdot \sqrt{d \cdot f}}. \tag{9}$$

The approximate distance based on the approximate mean acceleration is as follows:

$$dist_0 = \frac{1}{2} < a > t_{accel}^2 = \frac{1}{2} \frac{v_{limit}^2}{< a >} = \cdots = \frac{1}{a \cdot 4} \frac{m}{d} \ln \left( \frac{1+a}{1-a} \right). \tag{10}$$

This is opposed to the analytical distance, which is obtained by integrating the velocity, with the following results:

$$dist_1 = \int_0^{t_{accel}} v(t) dt = \cdots = \frac{m}{d} \ln \left( cosh \left( \frac{1}{2a} \ln \left( \frac{1+a}{1-a} \right) \right) \right). \tag{11}$$

By equating both distances, we obtain an expression for the parameter and with this, the optimal quadrature point. The equation is the following:

$$\frac{1}{a \cdot 4} \ln \left( \frac{1+a}{1-a} \right) = \ln \left( cosh \left( \frac{1}{2a} \ln \left( \frac{1+a}{1-a} \right) \right) \right) \tag{12}$$

whose numerical resolution obtains $a \sim 0.68$. Once estimated and with the kinematic magnitudes entered, we can obtain the expression for the average work as follows:

$$< W >= m < a > dist_0 = m \cdot \frac{2f}{m \ln\left(\frac{1+a}{1-a}\right)} \frac{a}{a \cdot 4} \cdot \frac{1}{d} \frac{m}{d} \ln\left(\frac{1+a}{1-a}\right) = \cdots = \frac{mf}{2d} \quad (13)$$

approximated from the rest up to $t_{accel}$. Note that this could be written as follows:

$$< W >= \frac{mf}{2d} = \frac{1}{2} m \left( \sqrt{\frac{f}{d}} \right)^2. \quad (14)$$

The kayak kinetic energy, when the limiting velocity $v_{limit} = \sqrt{f/d}$ is approximated, corresponds to the work required to accelerate the kayak from rest to that velocity, and is therefore independent of acceleration interval. The average power is written as follows:

$$< P >= \frac{W}{t_{accel}} = \frac{\frac{mf}{2d}}{t_{accel}} = \cdots = \frac{f}{\ln\left(\frac{1+a}{1-a}\right)} \cdot v_a. \quad (15)$$

### 2.4. Remarks and Quadrature Estimations

It is important to note that both the work and average power thus calculated approximate well to that which would be obtained numerically up to the instant $t_{accel}$. However, they somewhat underestimate the total time to complete the acceleration; $t_{accel}$ equals approximately the time to reach 85% of the asymptotic velocity. A simple alternative is to use the expression (7) to obtain a more approximate estimate of time, e.g., for $a = 0.99$, we obtain $t_{99} = \ln 199 \frac{m}{2f} \sqrt{\frac{f}{d}}$. In this way, we define an acceleration interval ($t_{accel}$) that allows us fast calculations of the acceleration and average power, and a stabilization interval up to ($t_{99}$) that will allow us to estimate the time or number of strokes to reach maximum speed, followed by a uniform speed interval (Figure 3). This description allows us approximate expressions with which to make quick calculations of the average time, distance, work and power. The area under the velocity–time curve gives us the distance (Figure 3) and the area under the curve $m \cdot a$ vs. $v \cdot t$ the work. Entering the parameters from Figure 1, we obtain that the paddler needs $t_{accel} = 8.59$ s to reach 85% of the asymptotic velocity and $t_{99} = 18.61$ s to complete the acceleration. The distance obtained is 20.19 m for 85% of the asymptotic velocity and 44.23 m for 99%. The average work and power obtained are the following:

$$W_{mean} = 992.65J, \qquad P_{mean} = 115.56W. \quad (16)$$

In the velocity versus time curve, the average acceleration is the slope of the secant straight by the point considered the end of the acceleration interval (see Figure 1). The area under the curve ($m \cdot a$) against distance ($v \cdot t$) represents the work we have seen, as the completed work is independent of the interval and only of the problem parameters and is identified with the kinetic energy of the kayak that has reached the asymptotic velocity.

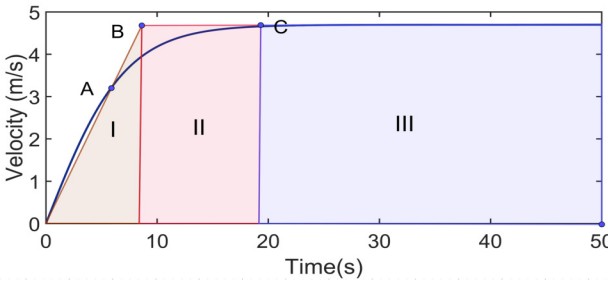

**Figure 3.** Velocity versus time curve, where the described intervals are drawn. (I) Constant acceleration interval, (II) stabilization interval and (III) constant velocity interval. (A) Point where 68% of velocity is reached, corresponding to time $t_{68}$, expression (7); (B) cut-off point between the (I) and (II) intervals, corresponding to time $t_{accel}$, expression (9); (C) cut-off points between intervals (II) and (III), corresponding to time $t_{99}$, expression (7). See text.

### 3. Analytical Solutions for Periodic Propulsion

The propulsion exerted by the paddler is periodic and alternating to the left and right, with a paddling frequency approximately between 50 strokes and 130 strokes per minute. It is possible to find studies that analyze the application of force in the stroke and its evolution with the frequency [8,9,16] as well as the relationship between stroke frequency and kayak speed [10,11]. There are also studies that obtained data of force or power applied on the paddle and their corresponding velocities [9,12,15,21].

The characteristics of the paddle and its dynamic effects are probably particular to this sport, and it is, therefore, of great interest to analyze them from a mathematical point of view. In what follows, we propose different models for this in an attempt to reproduce the characteristics of propulsion and to obtain a model that is as general as possible.

#### 3.1. Square Sine Stroke

One way to consider the periodic propulsion is modeling them with a trigonometric function. Hemon et al. [8] did this with a *sine* function but they only analyzed one stroke, which allowed them to avoid the negative value of this function in the second half of its period, which would represent the second stroke. This disadvantage forces us to use its absolute value and complicates the mathematical development considerably. Additionally, its time average is not favorable to our interests. Thus, in our case, we resort to a square sine function, $f = f_0 sin^2(wt)$, where $f_0$ is the force peak applied to the paddle, and $w = 2\pi \cdot \nu$ the angular frequency which accounts for the stroke frequency $\nu$. Our equation has the following form:

$$m\dot{v} + dv^2 = f_0 sin^2(wt) = \frac{f_0}{2}(1 - cos(2wt)). \tag{17}$$

This identity divides the force in two terms, on which there is a constant force of magnitude equal to half the peak force $\frac{f_0}{2}$ and another periodical force with twice the frequency of the original force. Let $f' = f_0/2$. We group the terms to obtain the following:

$$m\dot{v} = f'\left(1 - \frac{d}{f'}v^2\right) - f' \cdot cos(2wt). \tag{18}$$

We make the following change in variable:

$$u = \sqrt{\frac{d}{f'}} \cdot v \qquad \frac{du}{dt} = \sqrt{\frac{d}{f'}}\frac{dv}{dt} \tag{19}$$

and that leaves us with the following:

$$m\sqrt{\frac{f'}{d}}\dot{u} = f'(1 - u^2) - f' \cdot cos(2wt). \tag{20}$$

Let us now assume that the solution separates into the sum of two functions as follows:

$$u = g_1 + g_2 \qquad u^2 = g_1^2 + g_2^2 + 2g_1g_2 \qquad \dot{u} = \dot{g}_1 + \dot{g}_2. \tag{21}$$

Then, we have the following:

$$m\sqrt{\frac{f'}{d}}(\dot{g}_1 + \dot{g}_2) = f'(1 - g_1^2 - g_2^2 - 2g_1g_2) - f' \cdot cos(2wt). \tag{22}$$

Confronting the first term of the left member with the first two terms of the right member, we observe the following:

$$m\sqrt{\frac{f'}{d}}\dot{g}_1 = f'(1 - g_1^2). \tag{23}$$

That is an analogous equation to the solved one in the case of a constant force. Now, we have $f' = f_0/2$ and the solution for this part is the following:

$$g_1 = tanh\left(\sqrt{\frac{d}{f'}}\left(\frac{f'}{m}t + K'\right)\right). \tag{24}$$

Eliminating these three terms from the equation, we are left with the following:

$$m\sqrt{\frac{f'}{d}}\dot{g}_2 = -f'(g_2^2 + 2g_1g_2) - f' \cdot cos(2wt). \tag{25}$$

Here, we assume that the cross term $2g_1 \cdot g_2$ can be neglected. The resulting equation can be addressed by introducing the Fourier series from this point onward:

$$f(t) = a_0/2 + \sum_n a_n sin(w_n \cdot t) + \sum_n b_n cos(w_n \cdot t). \tag{26}$$

Restricted to the sine Fourier series, in this particular case we have the following:

$$g_2 = \sum a_n sin(w_n \cdot t) \qquad \dot{g}_2 \sim \sum a_n \cdot w_n \cdot cos(w_n t), \tag{27}$$

$$g_2^2 \sim \sum a_n^2 sen^2(w_n \cdot t) = \sum (a_n^2/2)(1 - cos(2w_n \cdot t)). \tag{28}$$

Since the proposed force has frequency $2w$, it is logical to assume that the solution has a strongly dominant term at the same frequency. Under this assumption, we can neglect the cross terms in $g_2^2$ and solve it in a much simpler way. We will come back to this in a later section. Assuming the above developments and using this strategy, the equation is written as follows:

$$m\sqrt{\frac{f'}{d}}\sum a_n \cdot w_n \cdot cos(w_n t) = -f'\sum (a_n^2/2)(1 - cos(2w_n \cdot t)) - f' \cdot cos(2wt). \tag{29}$$

By matching terms of the same frequency, we obtain the following coefficients:

$$
\begin{array}{lll}
n = 0 & \sum -\frac{a_n^2 f'}{2} = 0 & \frac{-a_2^2 f'}{2} + \frac{-a_4^2 f'}{2} + \frac{-a_6^2 f'}{2} + \cdots = 0 \\[2mm]
n = 1 & m\sqrt{f'/d} \cdot a_1 \cdot w = 0 & a_1 = 0 \\[2mm]
n = 2 & m\sqrt{f'/d} \cdot a_2 \cdot 2w = f'\frac{a_1^2}{2} - f' & a_2 = \frac{-f'}{m2w}\sqrt{d/f'} \\[2mm]
n = 3 & m\sqrt{f'/d} \cdot a_3 \cdot 3w = 0 & a_3 = 0 \\[2mm]
n = 4 & m\sqrt{f'/d} \cdot a_4 \cdot 4w = f'\frac{a_2^2}{2} & a_4 = \frac{f'}{2m4w}\sqrt{d/f'} \cdot a_2^2 \\[2mm]
n = 5 & \cdots & \cdots
\end{array}
$$

All the odd terms are null. We will build the solution with the 2 and 4 terms of the development.

$$g_2 = \frac{-f'}{m2w}\sqrt{\frac{d}{f'}}sin(2wt) + \frac{f'}{2m4w}\sqrt{\frac{d}{f'}}\left[\frac{f'd}{(m2w)^2}sin(4wt)\right]\cdots \tag{30}$$

The solution for $u$ is $u = g_1 + g_2$. Finally, we undo the change $u = \sqrt{d/f'} \cdot v$ to obtain $v = \sqrt{f'/d} \cdot u$, reaching the following result:

$$v = \sqrt{\frac{f'}{d}}tanh\left(\sqrt{\frac{d}{f'}}\left(\frac{f'}{m}t + K'\right)\right) - \frac{f'}{m2w} \cdot sin(2wt) + \frac{f'^2 \cdot d}{2m^3(2w)^24w} \cdot sin(4wt)\cdots \tag{31}$$

The first term represents the transient toward the limiting velocity and is analogous to the one we deduced in the constant force case. The remaining terms represent a periodic contribution that models the effect of the stroke. The first of these represents the dominant

contribution to the stroke frequency. The remaining ones are even terms of higher order and successively decreasing contribution; they are almost negligible in this case. Figure 4 represents the solution obtained, and in it, we observe both contributions: the transient and the periodic.

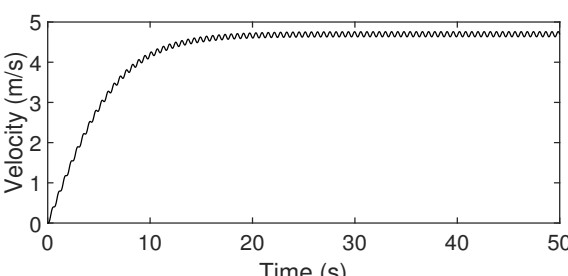

**Figure 4.** Kayak speed versus time, Square sine force case $m = 90$ kg, $f_0 = 120$ N, $d = 2.72$ kg/m y $w = 1.66 \cdot \pi$ rad/s (100 strokes/minute).

Returning to the transient, the value of the force is now replaced by the factor $f' = f_0/2$ and corresponds to the constant factor that appears when doubling the frequency in the functional expression of the force $f = f_0 sin^2(wt) = \frac{f_0}{2}(1 - cos(2wt))$. This constant term is equivalent to the time average of this expression for the force, which means that $< f_0 sin^2(wt) >= f_0/2$. Therefore, the limiting velocity becomes directly proportional to the root of the average force and inversely proportional to the drag coefficient.

$$v_{limit} = \sqrt{\frac{f_{ave}}{d}} \tag{32}$$

where $f_{ave}$ refers to the average force. It happens that this relationship between average force and speed has already been pointed out in canoeing field works [9], where the denomination of *area under the force curve* or *average force* is used. This will be one of the key factors in modeling a realistic paddling stroke, as we will discuss in the next and subsequent sections.

### 3.2. General Stroke Model

The velocity solution obtained for the sine–square stroke reproduces the expected behavior for a sprint kayak; at least, this is deduced from other analogous curves published in the literature [18,22], in this case for an Olympic rowing scull and a ergometer kayak, respectively. However, it is not able to reproduce the effect of stroke frequency over the speed. The stroke frequency only appears in the denominator of the oscillatory coefficients. It does not appear in the expression for the transient solution (31) and then, it has no influence over it. As commented, the time average of square–sine is independent of its angular frequency and of value 1/2. Thus, an increase in the stroke frequency while maintaining the peak force ($f_0$) will not have a reflection on speed; this is not a realistic behavior.

Field studies with competitive paddlers show a practically linear relationship between the stroke frequency and velocity [5,11]. These same studies show that as the stroke frequency increases, so does the maximum applied force and the ratio (average force)/(maximum force) [10], and as a consequence, also the velocity increases. Figure 5 and Table 1, both extracted from Gomes et al. [9], show the stroke dynamics from a field study with competitive paddlers. Figure 5 presents the force distribution versus time for different frequencies; Table 1 shows a whole series of magnitudes associated with it. These data show us the evolution of force peak, the duration of the aquatic phase, the aerial one and the mentioned ratio for the stroke impulse, which are valuable data to formulate a realistic stroke.

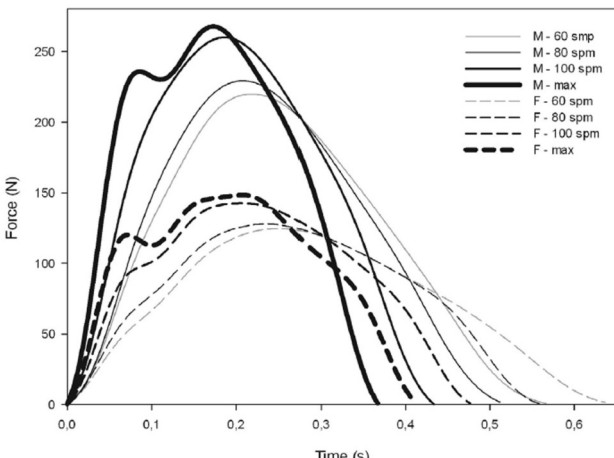

**Figure 5.** Average force–time curve for each gender and stroke frequency. Each curve represents the normalized mean curve (270, 300, 330 and 360 strokes from both sides at 60, 80, and 100 strokes per minute (spm) and at maximum frequency (max), respectively. M = men, F = women. Graph extracted from Gomes et al. [9].

**Table 1.** Table of mean $\pm$ SD (standard deviation) for the variables of the force–time curve for the canoeing stroke (Figure 5), for each gender and $F_{mean}/F_{max}$ for the whole sample. Extracted from Gomes et al. [9] and adapted.

| | Gender | Stroke Freq. [a] 60 | Stroke Freq. 80 | Stroke Freq. 100 | Stroke Freq. Race Pace |
|---|---|---|---|---|---|
| Frequency stroke (spm) | Men | $63 \pm 5$ | $81 \pm 3$ | $99 \pm 6$ | $124 \pm 7$ |
| | Women | $60 \pm 3$ | $79 \pm 6$ | $100 \pm 6$ | $112 \pm 3$ |
| Time in 200 m (s) | Men | $54.35 \pm 2.29$ | $47.85 \pm 2.00$ | $43.67 \pm 1.88$ | $38.68 \pm 0.83$ |
| | Women | $61.35 \pm 0.95$ | $53.76 \pm 1.73$ | $48.08 \pm 2.36$ | $44.94 \pm 1.21$ |
| Average velocity 200 m (m s$^{-1}$) | Men | $3.68 \pm 0.15$ | $4.18 \pm 0.18$ | $4.58 \pm 0.20$ | $5.17 \pm 0.11$ |
| | Women | $3.26 \pm 0.05$ | $3.72 \pm 0.12$ | $48.08 \pm 2.36$ | $44.94 \pm 1.21$ |
| Aquatic phase duration (s) | Men | $0.56 \pm 0.03$ | $0.50 \pm 0.04$ | $0.43 \pm 0.03$ | $0.37 \pm 0.03$ |
| | Women | $0.64 \pm 0.03$ | $0.55 \pm 0.02$ | $0.48 \pm 0.01$ | $0.43 \pm 0.02$ |
| Aerial phase duration (s) | Men | $0.40 \pm 0.05$ | $0.24 \pm 0.03$ | $0.18 \pm 0.04$ | $0.14 \pm 0.03$ |
| | Women | $0.35 \pm 0.01$ | $0.23 \pm 0.05$ | $0.17 \pm 0.05$ | $0.12 \pm 0.02$ |
| Time until $F_{max}$ (s) | Men | $0.22 \pm 0.03$ | $0.21 \pm 0.02$ | $0.19 \pm 0.02$ | $0.16 \pm 0.02$ |
| | Women | $0.26 \pm 0.03$ | $0.24 \pm 0.03$ | $0.21 \pm 0.02$ | $0.20 \pm 0.01$ |
| $F_{max}$ (N) | Men | $225 \pm 31$ | $234 \pm 32$ | $266 \pm 33$ | $274 \pm 35$ |
| | Women | $126 \pm 11$ | $130 \pm 8$ | $146 \pm 7$ | $153 \pm 11$ |
| $F_{ave}$ (N) | Men | $118 \pm 16$ | $128 \pm 18$ | $157 \pm 18$ [a] | $171 \pm 18$ |
| | Women | $72 \pm 6$ | $80 \pm 9$ | $92 \pm 13$ [a] | $99 \pm 15$ |
| Impulse (N·s) | Men | $66.3 \pm 7.3$ | $63.9 \pm 7.3$ | $67.7 \pm 9.5$ | $63.2 \pm 8.4$ |
| | Women | $46.5 \pm 5.9$ | $44.1 \pm 5.5$ | $44.2 \pm 6.3$ | $42.3 \pm 6.6$ |
| $F_{ave}/F_{max}$ (ratio %) | Entire sample | $53.3 \pm 3.3$ [c,d,e] | $57.2 \pm 3.9$ [b,d,e] | $61.0 \pm 3.8$ [b,c,e] | $64.8 \pm 3.7$ [b,c,d] |

Note. Men $n = 5$, Women $n = 5$. All of them analyzed for each stroke frequency, mean $\pm$ SD. spm = *strokes per minute*. [a] Stroke frequency (spm). [b] $p < 0.05$ significantly different to 60 spm. [c] $p < 0.05$ significantly different to 80 spm. [d] $p < 0.05$ significantly different to 100 spm. [e] $p < 0.05$ significantly different to race pace.

The analytical formalism that we presented is general enough to introduce new models for paddling. This is one of the big advantages of analytical model like this, in contrast to the pure numerical integrations of the differential equation. First, we introduce a model of stroke in the form of even power of the sine function and solve it analytically. Then, we use quadratic polynomial functions fitted to data from Gomes et al. [9] to try to approximate them to experimental data. The next step is to find a general mathematical model for the canoeing stroke in terms of time, and solve it analytically.

### 3.3. Stroke as Even Power of Sine

A more general stroke model is obtained by introducing the even powers of the sine function, that is, the following:

$$f(t) = f_0 sin^{2k}(wt) \qquad \text{with} \quad k = 1, 2, 3, 4, 5 \cdots \tag{33}$$

where for ($k = 1$), we recover the previous case. Now, as parameter $k$ takes discrete values successively greater, the associated *average force* is successively minor, and either an appropriate mathematical relationship or a correlation with experimental data would allow us to relate exponents and characteristic stroke frequencies. Figure 6 shows the *average force* for arbitrary values of $k$ and the functional form of those strokes (b).

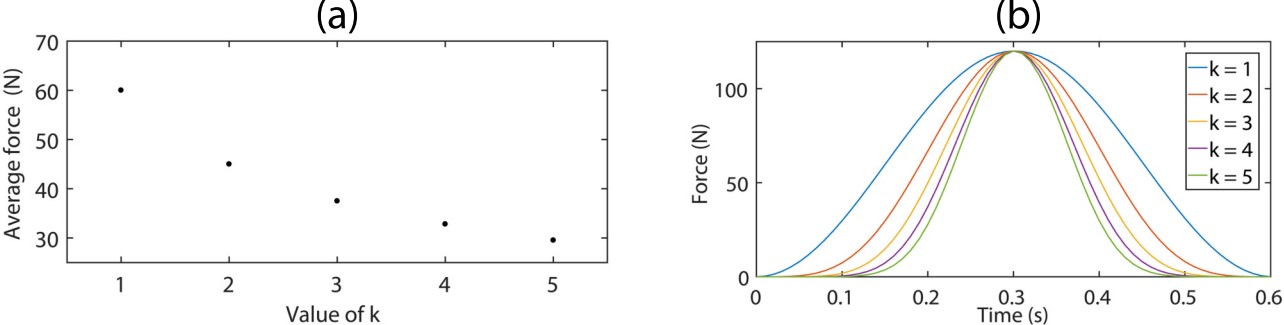

**Figure 6.** (**a**) Average force $\frac{f_0}{2^{2k}}\binom{2k}{k}$, of the stroke function (Equation (33)) for different values of parameter $k$ (see text), (**b**) stroke function (Equation (33)), with $f_0 = 120$ N, for the same values of parameter $k$ (see text).

Now, the equation for the kayak displacement takes the following form:

$$m\dot{v} + dv^2 = f_0 \cdot sin^{2k}(wt) \tag{34}$$

where the sine function could be developed according to the following expression:

$$sin^{2k}(wt) = \frac{1}{2^{2k}}\binom{2k}{k} + \frac{(-1)^k}{2^{2k-1}}[cos(2k \cdot wt) - \binom{2k}{1}cos((2k-2) \cdot wt) +$$
$$+ \cdots (-1)^{k-1}\binom{2k}{k-1}cos(2 \cdot wt)] \tag{35}$$

which converts the equation in the following expression:

$$m\dot{v} + dv^2 = \frac{f_0}{2^{2k}}\binom{2k}{k} + \frac{(-1)^k \cdot f_0}{2^{2k-1}}[cos(2k \cdot wt) - \binom{2k}{1}cos((2k-2) \cdot wt) +$$
$$+ \cdots (-1)^{k-1}\binom{2k}{k-1}cos(2 \cdot wt)]. \tag{36}$$

Again, we have a constant term that will serve as the average force. Now, $\frac{f_0}{2^{2k}}\binom{2k}{k}$ and a series of periodic terms are added to the oscillatory part of the solution. The analytical resolution is analogous to the previous case; the transient part of the solution is obtained with the stationary term; and the periodic part is solved by mean of a Fourier series in terms of sine. Due to the introduction of combinatorial numbers, it happens that the coefficients for the expression for sine raised to $2k$ differ for each value of $k$; thus, the periodic equation part does as well and it must be solved separately in each case. Table 2 shows such equations for the different values of $k$ solved.

**Table 2.** Equations obtained for each $k$ value.

| $k$ | Equation |
|---|---|
| 2 | $m\sqrt{\frac{f'}{d}}\sum_n a_n w_n cos(w_n t)=-f'\sum_n \frac{a_n^2}{2}(1-cos(2w_n t))+\frac{f_0}{2^3}[cos(4wt)-4cos(2wt)]$ |
| 3 | $m\sqrt{\frac{f'}{d}}\sum_n a_n w_n cos(w_n t)=-f'\sum_n \frac{a_n^2}{2}(1-cos(2w_n t))-\frac{f_0}{2^5}[cos(6wt)-6cos(4wt)+15cos(2wt)]$ |
| 4 | $m\sqrt{\frac{f'}{d}}\sum_n a_n w_n cos(w_n t)=-f'\sum_n \frac{a_n^2}{2}(1-cos(2w_n t))+\frac{f_0}{2^7}[cos(8wt)-8cos(6wt)+28cos(4wt)-56cos(2wt)]$ |
| 5 | $m\sqrt{\frac{f'}{d}}\sum_n a_n w_n cos(w_n t)=-f'\sum_n \frac{a_n^2}{2}(1-cos(2w_n t))-\frac{f_0}{2^9}[cos(10wt)-10cos(8wt)+45cos(6wt)-120cos(4wt)+210cos(2wt)]$ |

Table 3 shows the coefficients for each of these equations, and in this case, all the odd terms are null. The coefficients are written taking into account the change of variable described in the previous section $u = \sqrt{d/f'}\cdot v$ or $v = \sqrt{f'/d}\cdot u$ so that the coefficient $\sqrt{f'/d}$ appearing in the equation's left-hand part of Table 2 is removed. Writing the solution in its general form, we have the following:

$$v = \sqrt{\frac{f'}{d}}tanh\left(\sqrt{\frac{d}{f'}}\left(\frac{f'}{m}t+K'\right)\right) + a_2 \cdot sin(2wt) + a_4 \cdot sin(4wt) + a_6 \cdot sin(6wt)+ \\ +a_8 \cdot sin(8wt) + a_{10} \cdot sin(10wt) + a_{12} \cdot sin(12wt)\cdots \quad (37)$$

where now, $f' = \frac{f_0}{2^{2k}}\binom{2k}{k}$, and the coefficients refer to the Table 3. Figure 7 shows the velocity curves for the different values of $k$.

**Table 3.** Table with the obtained coefficients.

| $k$ | $a_2$ | $a_4$ | $a_6$ | $a_8$ | $a_{10}$ | $a_{12}$ |
|---|---|---|---|---|---|---|
| 2 | $\frac{-f_0}{2mw_2}$ | $\frac{1}{mw_4}\left[\frac{f'a_2^2}{2}+\frac{f_0}{8}\right]$ | $0$ | $\frac{f'}{2mw_8}a_4^2$ | $0$ | $0$ |
| 3 | $\frac{-15f_0}{2^5 mw_2}$ | $\frac{1}{mw_4}\left[\frac{f'a_2^2}{2}+\frac{6f_0}{2^5}\right]$ | $\frac{-f_0}{2^5 mw_6}$ | $\frac{f'}{2mw_8}a_4^2$ | $0$ | $0$ |
| 4 | $\frac{-56f_0}{2^7 mw_2}$ | $\frac{1}{mw_4}\left[\frac{f'a_2^2}{2}+\frac{28f_0}{2^7}\right]$ | $\frac{-8f_0}{2^7 mw_6}$ | $\frac{f'}{mw_8}\left[\frac{f'a_4^2}{2}+\frac{f_0}{2^7}\right]$ | $0$ | $\frac{f'}{2mw_{12}}a_6^2$ |
| 5 | $\frac{-210f_0}{2^9 mw_2}$ | $\frac{1}{mw_4}\left[\frac{f'a_2^2}{2}+\frac{120f_0}{2^9}\right]$ | $\frac{-45f_0}{2^9 mw_6}$ | $\frac{f'}{mw_8}\left[\frac{f'a_4^2}{2}+\frac{10f_0}{2^9}\right]$ | $\frac{-f_0}{2^9 mw_{10}}$ | $\frac{f'}{2mw_{12}}a_6^2$ |

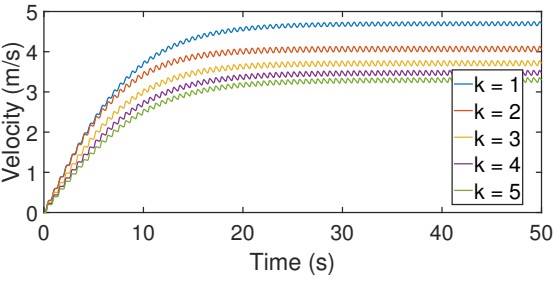

**Figure 7.** Velocity versus time with $m = 90$ kg, $f_0 = 120$ N, $d = 2.72$ kg/m y $w = 1.66\cdot\pi$ rad/s (100 strokes/minute), for the stroke function $f = f_0 sin^{2k}(wt)$ and different values of the parameter $k$ (see text).

### 3.4. Stroke as Quadratic Function

Average force over the stroke is the main magnitude that affects the final velocity. The amplitude of the oscillatory part is inversely proportional to the mass, to the main frequency and its multiples. As commented, field studies show that the stroke frequency has a linear

relation with velocity and it is also known that both peaks, as mean forces, increase as the frequency does; this relation is not correctly reflected in the solved expression. The function proposed in the previous section, $f(t) = sin^{2k}(wt)$, only allows some flexibility but has two inconveniences. The first refers to the fact that only discrete values of the parameter are allowed, and it must be fit to different characteristic frequencies. The second is the fact that the greater force average is of $1/2$ value and corresponds to the lower parameter $k = 1$, being lower than the other averages. Table 1 of stroke magnitudes, extracted from Gomes et al. [9], shows an average force of $53.3 \pm 3.3\%$ over the maximum force for a stroke frequency of 60 strokes per minute, being that this average is greater for higher frequencies. This leads us to assume that the function $f(t) = sin^{2k}(wt)$, with $k = 1$, may be correct for modeling that frequency, but the higher parameters would only be suitable for lower stroke frequencies.

Higher frequencies must be modeled by a different functional relation. An alternative is the polynomial fit, quadratic in its simpler version; it allows us to reproduce the searched stroke parameters, the force maximum and the aquatic and aerial phase duration. Using such a function, the average force is equal to 66.66% of its peak value, and is, therefore, particularly well-suited to fit the maximum stroke frequency curve. Using functions of this kind and resorting to values of peak force and aquatic and aerial phase duration for maximum frequencies from Table 1, we define piecewise functions to model the men and women's stroke. Their mathematical definition is as follows:

$$f_M(t) = \begin{cases} -8005.84t^2 + 2962.16t & t \in (0, 0.37) \\ 0 & t \in (0.37, 0.51) \end{cases} \tag{38}$$

in the case of men, and

$$f_W(t) = \begin{cases} -3309.90t^2 + 1423.26t & t \in (0, 0.43) \\ 0 & t \in (0.43, 0.55) \end{cases} \tag{39}$$

in the women's case. These functions are not periodic, but if we write them in terms of a Fourier series in sine and cosine, we can use the ideas that we have developed. The general Fourier series are introduced in Equation (26), and written with uppercase coefficients here to be coherent with the notation that follows.

$$f(t) = A_0/2 + \sum_n A_n sin(w_n \cdot t) + \sum_n B_n cos(w_n \cdot t). \tag{40}$$

Tables 4 and 5 show the coefficients for the six first terms in sine and cosine of both functions.

**Table 4.** Fourier series coefficients for $f_M(t)$.

| $A_0$ | $A_1$ | $A_2$ | $A_3$ | $A_4$ | $A_5$ | $A_6$ |
|---|---|---|---|---|---|---|
| 265.05 | −97.96 | 0.37 | −11.18 | −8.99 | −1.12 | −0.72 |
| – | $B_1$ | $B_2$ | $B_3$ | $B_4$ | $B_5$ | $B_6$ |
| – | 114.36 | 2.40 | −6.92 | 2.86 | 2.65 | −1.45 |

**Table 5.** Fourier series coefficients for $f_W(t)$.

| $A_0$ | $A_1$ | $A_2$ | $A_3$ | $A_4$ | $A_5$ | $A_6$ |
|---|---|---|---|---|---|---|
| 159.49 | −63.36 | −1.57 | −1.43 | −4.02 | −2.99 | −0.77 |
| – | $B_1$ | $B_2$ | $B_3$ | $B_4$ | $B_5$ | $B_6$ |
| – | 51.81 | 7.74 | −2.70 | −1.70 | 0.88 | 1.13 |

Figure 8 shows the piecewise function overlapped with the Fourier series that approximates it. The differential equation, writing the force in terms of a Fourier sine and cosine development, is as follows:

$$
\begin{aligned}
m\dot{v} + dv^2 = A_0/2 + A_1 sin(wt) + A_2 sin(2wt) + \cdots + A_6 sin(6wt) + \\
+ B_1 cos(wt) + \cdots + B_6 cos(6wt).
\end{aligned}
\tag{41}
$$

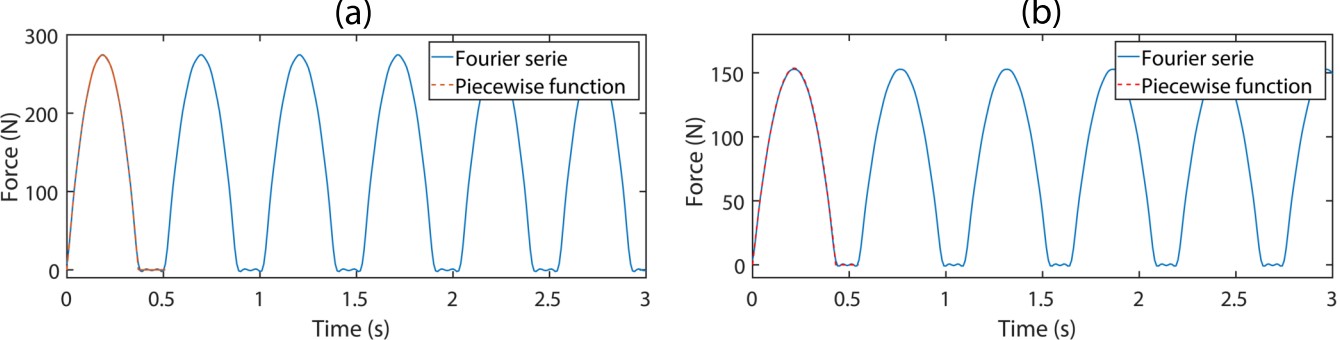

**Figure 8.** (**a**) Piecewise function for men's category, defined by expression (38) and representative Fourier series. (**b**) Piecewise function for women's category, defined by expression (39) and representative Fourier series.

The procedure is analogous to the previous equations. Variable changes and simplifications analogous to the previous cases are made, the constant term $A_0/2$ contributes to the transient solution, and the periodic terms contribute to the oscillatory one. We develop the periodic velocity in terms of sine and cosine, and rewrite the equation to arrive at the following:

$$
\begin{aligned}
m\sqrt{\frac{f'}{d}} \sum_n (a_n w_n cos(w_n t) - b_n w_n sin(w_n t)) + f'(\sum_n a_n sin(w_n t) + b_n cos(w_n t))^2 = \\
= \sum_{n=1}^{6} (A_n sin(w_n t) + B_n cos(w_n t)).
\end{aligned}
\tag{42}
$$

It happens again that the quadratic factor, written in the left part of the equation, introduces crossed terms that complicate the resolution. Assuming that the $A_1$ and $B_1$ terms are dominant in the proposed stroke model, Tables 4 and 5, we can neglect the rest of the crossed terms and consider only the contribution of the $2a_1 b_1$ term to the equation. This approximation allows us to solve analytically and estimate its overall contribution to the result. Expanding the square term, we have the following:

$$
\begin{aligned}
m\sqrt{\frac{f'}{d}} \sum_n (a_n w_n cos(w_n t) - b_n w_n sin(w_n t)) = -f'\left(a_1^2 sin^2(wt) + b_1^2 cos^2(wt) + a_1 b_1 sin(2wt)\right) + \\
+ \sum_{n=1}^{6} (A_n sin(w_n t) + B_n cos(w_n t))
\end{aligned}
\tag{43}
$$

where we have simplified the crossed term at double frequency $2a_1 b_1 sin(wt)cos(wt) = a_1 b_1 sin(2wt)$, which is precisely the crossed term mentioned above. We group terms and rewrite them to arrive at the following:

$$
\begin{aligned}
m\sqrt{\frac{f'}{d}} \sum_n (a_n w_n cos(w_n t) - b_n w_n sin(w_n t)) = \\
= -f'\left(\frac{a_1^2 + b_1^2}{2} + \frac{a_1^2 - b_1^2}{2} cos(2wt) + a_1 b_1 sin(2wt)\right) + \sum_{n=1}^{6} (A_n sin(w_n t) + B_n cos(w_n t)).
\end{aligned}
\tag{44}
$$

the expression from which we can equal terms of the same frequency to arrive at the following:

$$
\begin{array}{llll}
n = 0 & f'(\frac{a_1^2 + b_1^2}{2}) = 0 & -- \\
n = 1 & mwa_1 = B_1 & -mwb_1 = A_1 \\
n = 2 & m2wa_2 = B_2 + \frac{f'}{2}(a_1^2 - b_1^2) & -m2wb_2 + f'a_1b_1 = A_2 \\
n = 3 & m3wa_3 = B_3 & -m3wb_3 = A_3 \\
n = 4 & m4wa_4 = B_4 + \frac{f'}{2}(a_2^2 - b_2^2) & -m4wb_4 = A_4 \\
n = 5 & m5wa_5 = B_5 & -m5wb_5 = A_5 \\
n = 6 & m6wa_6 = B_6 + \frac{f'}{2}(a_3^2 - b_3^2) & -m6wb_6 = A_6
\end{array}
\tag{45}
$$

Thus, we arrive at the following general solution:

$$
v = \sqrt{\frac{f'}{d}} \tanh\left( \sqrt{\frac{d}{f'}} \left( \frac{f'}{m} t + K' \right) \right) + \sum_{n=1}^{6} (a_n \cdot sin(nwt) + b_n \cdot cos(nwt))
\tag{46}
$$

where now $f' = A_0/2$ and the coefficients $a_n$ and $b_n$ are cleared from the previous expressions. With the intention of evaluating the precision of the approximation assumed in the analytical resolution, in Figure 9, we represent this solution for men and women against the numerical resolution of the differential equation in both cases (see the detail of the figures). It is observed that the result is quite approximate, as only a small displacement appears at the beginning of the transient and especially in men. The weight of the crossed term $f'a_1b_1$ on $A_2$ (see row $n = 2$ in the expressions (45)) is estimated at 6.6% for men and less than 1% for women; therefore, its influence on the solution is limited. In any case, it is observed that the agreement between the numerical and analytical solutions is quite good in both cases.

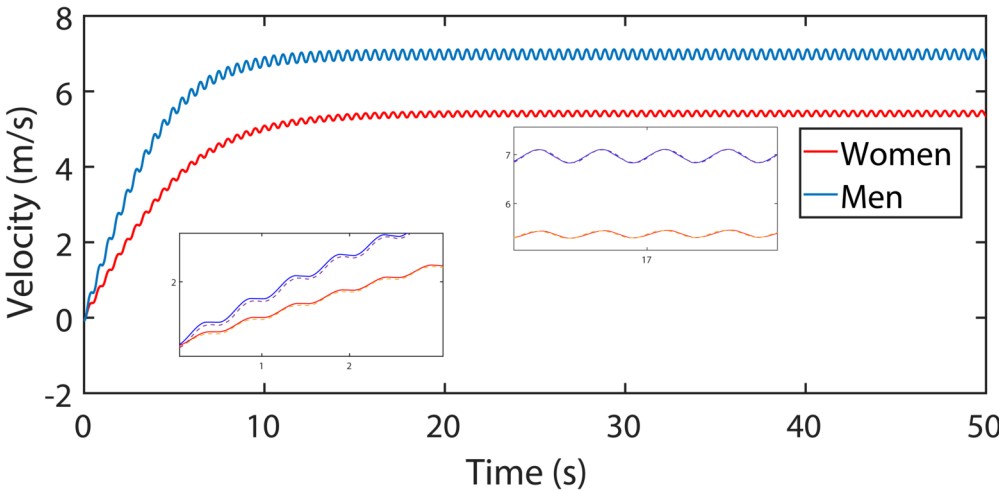

**Figure 9.** Analytical solution for maximum frequency and quadratic stroke: women and men. In detail, the comparison with the numerical solution (in dashed line) for both cases and both in the initial transient and in the final part of the curves.

The characteristics of the represented solutions in Figure 9 are again given by the stroke model used. The stroke frequency is about 109 strokes/minute for women and 117 strokes/minute for men. The limit velocity is given by $v_{limit} = \sqrt{f'/d}$, about 6.98 m/s in men and about 5.41 m/s in women. In addition, the value of $f' = A_0/2$ fits quite well the value of average force, which could be estimated for the Gomes table in Table 1. For instance, we find a peak force for women of 153N, which, modeled by a quadratic function and taking the relative weight of aquatic versus aerial phases, gives $153 \cdot 0.666 \cdot 0.781 = 79.58$ N, which is a very close approximation of the value $f' = A_0/2 = 79.746$ N obtained from

Table 5 for women. A quite similar result can be obtained for men. Finally, the stroke frequency appears explicitly in the coefficients for the amplitudes of the oscillatory part of the solution and implicitly through the stroke model, which is fitted for the competition frequency and solved analytically for it.

## 4. Estimations of the Model

Although we do not have our own field data, we could resort to the literature to analyze the model predictions. Let us first discuss some kinematic estimations based on the quadrature proposed and followed by estimations of drag, velocity and power, the last also being based in quadrature. After that, fittings of both asymptotic and whole curves to the literature data are made and discussed.

### 4.1. Kinematic Estimations

Many estimations can be made from the approximate expression described in Section 2.3 and represented in Figure 3. With reference to Figure 1 and with its numerical values ($m = 90$ kg, $f = 60$ N, $d = 2.72$ kg/m and $w = 1.66 \cdot \pi$ rad/s (100 strokes/minute)), we obtain an acceleration interval of $t_{accel} = 8.59$ s and stabilization of $t_{99} = 18.61$ s to complete the total acceleration interval. The distances corresponding to these times are 20.19 mts and 44.23 mts, respectively. If we increase the drag coefficient to $d = 4.0$ kg/m, these percentages of the limiting velocity correspond to the following times of $t_{accel} = 7.08$ s and $t_{99} = 15.37$ s, and distances of 13.72 mts and 30.08 mts, respectively. For a frequency of 100 strokes per minute, it corresponds to one stroke every 0.6 s. Therefore, according to this estimation, for $d = 2.72$ kg/m, the paddler takes about 31 strokes to reach 99% of his speed, and for $d = 4.0$ kg/m, it will take slightly less, about 26 strokes. This is practically the same number of strokes to reach the maximum speed reported by Treus et al. [15] in a study of accelerometry applied to canoeing.

### 4.2. Drag Coefficient and Limiting Speed Estimations

Studies of applied force versus speed in real paddlers allow us to analyze the limiting speed. Initially, we use the above-mentioned coefficient $d = 2.72$ kg/m, extracted from a field study [4]. It is analogous and of the same order as other hydrodynamic studies at constant speed and propulsion [1–3,6], but it is a passive hydrodynamic drag coefficient. From the data of Gomes et al. [9], for the competition paddling frequency, we have peak forces of 274 N and 153 N, for male and female categories, respectively, which transform to average forces of $274 \cdot 0.666 \cdot 0.726 = 132.48$ N and $153 \cdot 0.666 \cdot 0.782 = 79.68$ N, respectively, and where we have taken into account both the aquatic and aerial phases (factor 0.726 in men and 0.782 in women), as detailed in the previous Section 3.4. The expression for the limiting velocity allows us to either obtain velocities from a known $d$ coefficient or to estimate the drag coefficient from the velocity data obtained in the study. Using the expression (32) and the passive resistance coefficient $d = 2.72$ kg/m, we obtain velocities of $v_M = 6.98$ m/s and $v_W = 5.41$ m/s for males and females, respectively, appreciably above the average velocities obtained in the study $v_M = 5.17$ m/s and $v_W = 4.45$ m/s. Similar results can be obtained for the rest of the data at lower stroke frequencies. Regarding the kinematic–dynamic study over 200 m by Treus et al. [15], the conclusions are similar. From their data, we take the fastest interval (between 50 and 100 m) corresponding to a force of 276.31 N and average velocity of 5.2 m/s for the male category. Such force, averaged as before and with a coefficient of $d = 2.72$ kg/m, can lead us to a much higher velocity, of value $v = 7.01$ m/s, analogous to those obtained for Gomes et al. [9]. Similar results can also be obtained from more recent data [10]. If, on the other hand, we extract drag coefficients from field data, we obtain $d = 4.96$ kg/m and $d = 4.02$ kg/m for males and females, respectively, from the data of Gomes et al. [9] and $d = 4.94$ kg/m from that of Treus et al. [15], being of the same order for the rest of the data or stroke frequencies and relatively constant by gender. We can evaluate the dynamic magnitudes from these force and drag coefficient estimates. We transfer the data to the expressions (16). Using

the remaining ones in Figure 1, we obtain work to complete the acceleration interval of $W_M = 1201.94$ J and $W_W = 891.94$ J and their corresponding estimated average power at $P_M = 280.77$ W and $P_W = 145.47$ W for males and females, respectively.

We have no data and cannot make any statements about the efficiency of the stroke. Additionally, the forces were obtained or correlated with accelerometry data measured in situ in the field studies. Therefore, the observed differences must have their origin in the simplifications assumed, which ultimately refer to the differences between active and passive hydrodynamic drag. From Pendergast's studies of human locomotion in water [23], it follows that active hydrodynamic drag is far superior to passive drag in all forms of human locomotion of this type. In particular for sprint canoeing, an estimate is made for active drag of the type $f_A = 3.2 \cdot v^{2.46}$ with the particularity of being a *free* exponent expression sometimes used for hydrodynamic drag adjustments [1]. A *phenomenological* estimate tells us that in these settings, *higher exponents correspond to lower coefficients*, and the equivalent expression with squared exponent corresponds to a considerably higher coefficient. For example, for velocities of 5.17 and 4.45 m/s, the active hydrodynamic drag for the above expression is 181.10 and 125.93 N and corresponds to coefficients of $d = 6.81$ and $d = 6.36$ kg/m if we use quadratic-type hydrodynamic drag expressions. We see that these are high estimates and well above the various passive hydrodynamic drag coefficients obtained in the literature. On the other hand, the curves obtained from the model suggest that the assumptions made and the resolution are not far fetched, as it is not difficult to find similar curves in the literature—for example, in the work of Leroyer [17] by numerical calculation, in Begon's [18] where a kayak ergometer is modeled, or in Labbé [22] or Buckmann's [24] works applied to Olympic rowing.

### 4.3. Asymptotic Curve Fitting

It is also possible to adjust speed–distance curves published by other authors. For example, Pickett et al. [25] present speed–distance curves over 200 m with elite paddlers. These curves show the acceleration interval, the passage through a speed maximum and a third phase of slight deceleration associated with fatigue. The data are obtained with an interval of 10 m and are not precise enough to observe the dynamics of each stroke, but they allow estimations of the asymptotic term. As our model does not consider fatigue, we assume that the maximum velocity corresponds to 99% of the asymptotic velocity and that, thereafter, no fatigue appears, i.e., the velocity is maintained or increases very slightly toward its asymptotic velocity. As the data were measured with elite paddlers, we also assume that their average force is of the order of those already published by Gomes et al. [9] or Treus et al. [15], and therefore are of the order of $f_{ave} = 130$ N. From the expression obtained in Equation (6), and with the assumptions made, we can clear the resistance coefficient.

$$ d = \frac{f_{ave}}{v_a/a} = \frac{130}{5.78/0.99} = 3.81 \text{ kg/m}. $$

and then we obtain the mass from the integrated expression for the distance, Equation (11). From the Pickett data, the distance where the 99% of velocity is reached is 50 m. Therefore, by clearing in the expression, we find the following:

$$ m = \frac{50 \cdot d}{\ln\left(cosh\left(\frac{1}{2a} \cdot \ln\left(\frac{1+a}{1-a}\right)\right)\right)} = \cdots = 96.01 \text{ kg}. $$

As we can see, both the resistance coefficient and the mass obtained offer reasonable values. Now, we can transfer these four parameters to the transient expression to compare on a graph experimental versus estimated values. The general expression for the transient is written as follows:

$$ v = \sqrt{\frac{f}{d}} tanh\left(\sqrt{\frac{d}{f}}\left(\frac{f}{m}t\right)\right). $$

With the integration constant $K = 0$ if the starting is from rest. However, to compare, we must replace time with abscissa distance. Its equivalence in our model can be obtained by integrating as in Equation (11) to obtain the following:

$$dist = \frac{m}{d}\left[ \ln\left( cosh\left( \sqrt{\frac{d}{f}} \cdot \frac{f}{m} \cdot t \right) \right) \right].$$

With these data, the representation is presented in Figure 10. This adjustment is probably no more than an estimate based on reasonable assumptions, but although the first experimental point is not well reproduced, on the whole, it provides realistic results and the curve fits quite well the rest of the points. It must be taken into account that the oscillations due to strokes are specially relevant in the first part of acceleration and they are not recorded in the experimental data, so a significant noise amount must be supposed. Note that one could also proceed by a multiparameter fit ($f$, $d$, $m$) by minimization. In fact, by this method, one obtains curves very close to the experimental one but with unrealistic parameters. Moreover, several combinations of parameters are possible, which is not consistent and shows that our fitting strategy for this estimation is more appropriate. Note also that a more appropriate fit could be made if at least some of the parameters are previously known.

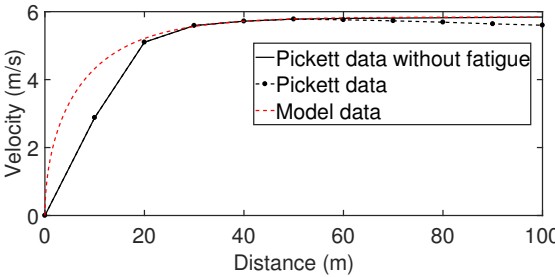

**Figure 10.** Velocity versus distance curves, where Pickett et al. [25] field data can be compared with those obtained by the model. See text for details.

Finally, if we resort to the optimal quadrature point (from Equation (12)), of value $a \sim 0.68$, the estimated work and acceleration power for this curve are $W = 1525.4$ J and $P = 311.4$ W, respectively—higher values but not far from those obtained for the estimations of Gomes et al. [9] or Treus et al. [15].

### 4.4. Whole Curve Fitting

Additionally, Bonaiuto et al. [12], using a multichannel digital acquisition (DAQ) system, obtained synchronous field data of stroke force, velocity and displacement that could be also fitted using our model. These experimental data reflex the intra-stroke dynamics. We begin modeling the stroke by using a piecewise function quadratic for the aquatic phase and null for the aerial one, from which we extract the Fourier sine and cosine coefficients (Table 6); both the piecewise function and Fourier series are represented in Figure 11.

**Table 6.** First Fourier series coefficients for the paper stroke.

| $A_0$ | $A_1$ | $A_2$ | $A_3$ | $A_4$ | $A_5$ | $A_6$ |
|---|---|---|---|---|---|---|
| 100.048 | −0.0371 | −30.3862 | −0.0037 | −7.5965 | −0.0013 | −3.3762 |
| – | $B_1$ | $B_2$ | $B_3$ | $B_4$ | $B_5$ | $B_6$ |
| – | 77.4296 | −0.0255 | 2.8831 | −0.0130 | 0.6294 | −0.0086 |

The average force is about $150 \cdot 0.5 \cdot 0.667 = 50$ N, and the equilibrium velocity can be extracted from the paper graph. We take $v = 2.714$ m/s; from both drag coefficients, we

can estimate $v = \sqrt{\frac{f_{ave}}{d}} \rightarrow d = \frac{f_{ave}}{v^2} = 6.786$ kg/m. The estimated mass is about 80 kg (the paddler mass is 68 kg). Finally, the stroke frequency is over 65 strokes/minute and can be easily estimated from the paper figure; we fine tune its exact value to better reproduce the oscillatory part of the solution to conclude with a value of 64.95 strokes/minute. These are the set of parameters needed to feed the model. By using it and the stroke model described, the differential equation can be solved following the procedure of Section 3.4 and compared with the published graph. Figure 11 shows this comparison.

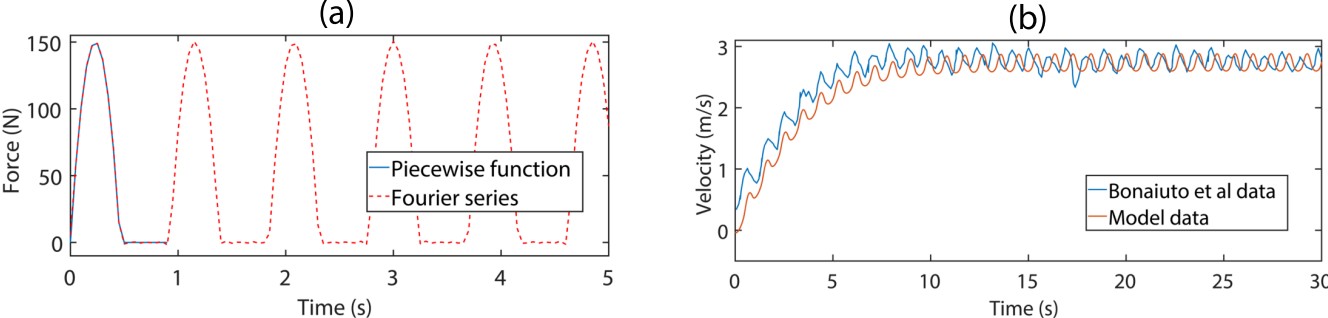

**Figure 11.** (**a**) Stroke model used, piecewise function versus Fourier series approximation, and (**b**) Bonaiuto et al. [12] experimental data versus model data; horizontal axis is time in seconds. See text for details.

It is shown that, as both velocity oscillation and transient evolution are well reproduced and although the analytical solution is slightly below the experimental one, this is related to instabilities in the experimental velocity in the second part of the graph since they condition the parametrization.

## 5. Discussion of the Model

We have seen how the problem of the propulsion of a kayak, formulated in the proposed mathematical terms, always shows two components, a transient and a periodic one. The first is a consequence of the appearance of a constant force term, which is generally identified with the zeroth order term of the Fourier development of the stroke function, while the second is due to the characteristic periodic propulsion of kayaks, canoes or rowing boats.

The transient term has two remarkable features: a concave interval below acceleration starting from rest and a horizontal asymptote in the limit of long times. The acceleration interval is modulated by the coefficient that multiplies the time at the hyperbolic tangent, is directly proportional to the root of the average force and the drag coefficient and is inversely proportional to the mass. Regarding the asymptote, it allows us to extract the equilibrium, or limit, velocity in terms of the average force and drag coefficient. It is interesting to note again how the mass does not appear in the expression for the equilibrium velocity and its influence is inverse in the transient toward it, that is, the greater the mass, the slower the kayak accelerates toward the equilibrium velocity, although its final value is independent of it. The remaining factors, the average force and the drag coefficient, appear in both terms.

The oscillatory term is superimposed on the transient and is directly determined by the stroke properties. In this paper, we have modeled three variants: a squared sine presenting a constant factor and a single frequency: even power sine-type strokes, with an even term development on cosine; and the quadratic stroke, which is not a periodic function but can be approximated well with sine and cosine series. Any stroke model, no matter how complex, must have a dominant frequency, which corresponds to the stroke frequency and a series of subsidiaries, which account for the functional details of it—in our case, up to 13 coefficients to model the quadratic stroke. The coefficients of the oscillatory part account for the amplitude of the oscillatory solution.

They modulate the oscillation of the velocity, and hence, the acceleration, over the transient curve and are inversely proportional to the mass, the stroke frequency and the powers of both in the higher order terms. Their amplitude can be estimated from the coefficients of the analytical solutions; in the case of polynomial stroke, as modeled in Section 3.4, an oscillation amplitude over the limiting velocity of 3.7% and 2.9% is obtained for males and females, respectively, but it could be much higher if the stroke frequency or mass are lower, as shown in Figure 11, with an approximate amplitude of 10%. The mass accounts for its effect on acceleration since mass is the parameter that multiplies acceleration on the equation; the greater the mass, the smaller the amplitude and the smaller the velocity oscillations. With the frequency, something analogous happens: the higher the frequency, the more strokes are made, and with more strokes, we get closer to the situation of constant propulsion. Therefore, the amplitude of the oscillations is reduced and vice versa.

Finally, perhaps the main unfinished business is the formulation of a general stroke model that, under a single mathematical formulation, allows modeling of all stroke parameters. These parameters are peak force, average force, as well as accounting for the full range of stroke frequencies, and even the functional details of stroke. Some of these features we have introduced by using even periodic functions or piecewise polynomial ones. We have shown how the analytical resolution decomposes the stroke function into a linear combination of its Fourier components, how the zero frequency coefficient allows the resolution of the asymptotic part, and how the remaining ones model the periodic part. This resolution strategy demonstrates that a more general stroke model is solvable, an intermediate average force (between the sine-square and the quadratic polynomial) can be modeled by a linear combination of both, or the occurrence of submaximals (Figure 4) can be modeled by the linear combination of periodic or polynomial functions whose maxima are time shifted. All these cases are analytically solvable. This strategy is also applicable with respect to the stroke frequency, that is, a linear combination of a periodic function, for example, a sine-square, and a quadratic could be formulated to be a piecewise function that accounts for the water and air phases of the paddling. It would even be feasible to make a general formulation that accounts for the expected range of paddling frequencies. Although analytical resolution are feasible, analogous to Section 3.4, it is not possible to extract simple mathematical expressions in general beyond expressing them in terms of the Fourier coefficients of the stroke function, which have to be particularized for each case.

## 6. Conclusions

In this paper, we introduced a simple analytical model for the fluid–hull–paddlers system, which allows to understand the basic physics of the problem and discuss many relevant aspects of the race in simple and useful terms. We proposed three models for the periodic force: a squared sine with a single frequency, a force made of even powers of sine-type or cosine strokes, and a quadratic stroke approximated by a periodic series.

We have shown that the solution for these system can always be presented as an asymptotic term and a series of periodic terms. The asymptotic term corresponds to the terminal velocity, which depends basically on the average force, the drag coefficient and the periodic terms, which depend on the properties of the strokes.

This general solution allow us to define quadrature relations, such as the average acceleration, acceleration time, the applied work or the average power in terms of simple physical constants. These quantities are useful because they connect the physics of the system to quantities that are useful to make analysis of the race and the performance of the athlete in simple terms. Additionally, these expressions allow to observe the importance of the hydrodynamics properties of the system and those related to the performance of the athlete as the average force or the frequency.

We compared these models to data from the literature and found good correspondence with our model, which allows us to discuss in detail many issues of the race. This means that the one-dimensional models of the kayak captures the most important aspect of the

system and provides a bridge between basic physics principles to important aspects of the hydrodynamics, the race and the performance of the sport.

The model provides a important tool to analyze the performance of the race and develop strategies in a large number of scenarios to improve performance. The value of these analytical models is that they provide simple and meaningful expressions, which can be adapted to many situations of the race.

In future works, we will explore the possibility to produce even better models of the stroke based on data and to include other aspects of the race to test their importance in overall performance.

**Author Contributions:** The mathematical model, its development and paper structure were completed by D.D. The first draft was written by D.D. The paper and math were revised by C.R. The quadrature section was proposed by C.R. and implemented by both authors. C.R. wrote part of the abstract and introduction and the full conclusion. Both authors contributed to the final paper revision, including figures and comparisons with published data. All authors have read and agreed to the published version of the manuscript.

**Funding:** This research received no external funding.

**Institutional Review Board Statement:** Not applicable.

**Informed Consent Statement:** Not applicable.

**Data Availability Statement:** The data that support the findings of this study are available from the corresponding author, upon reasonable request.

**Conflicts of Interest:** The authors declare no conflict of interest.

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
