# Peer review of "One-Dimensional Mathematical Model for Kayak Propulsion"

_applsci, doi:10.3390/app112110393_

Round 1

Reviewer 1 Report

Dear Authors,

Thank you for the opportunity to review your interesting paper. However, I would recommend some changes in the manuscript.

  1. I find a lack of clearly formulated aim (purpose) of the paper.
  2. The re-organization of the article's sections could, in my opinion, increase its transparency. For example, section 2 Mathematical model is composed of: introduction of the main model, Equation (1), direct solution of this model for constant force, approximate solution (quadrature) for constant force and discussion in lines 136-157. Would dividing into more subsections concerning these issues increase readability of this section? I think so.
  3. Analytical solution of Equation (2) given, for constant force, by Equation (3) results obviously from the well-known formula for derivative of the hyperbolic tangent. A brief comment or reference to the literature would be advisable.
  4. Equation (6) – why is “/” used instead of, e.g., “with”, as in equation (3)?
  5. Figures 1 and 2 refer to the constant force case; angular frequency ω  in the description is superfluous.
  6. Why in the title of subsection 2.1 and especially in line 106 the term "asymptotic solution" is used for the solution given by Equation (3)? This term, which would later be justified in the case of periodic force, seems premature here.
  7. Line 125:“The traveled distance up to that moment will be” – next the approximate distance dist0  described by Equation (10) is given. This formulation seems ambiguous. Consider clear “approximate distance based on the approximate mean acceleration <a>”, the more so that in line 126, after equation (10), the distance traveled calculated on the basis of the model (3) is called “the analytical distance”. More precise wording would be desirable, also elsewhere.
  8. Line 130 – “graph resolution” - not a numerical solution?
  9. Equation (13) - shouldn't be exactly dist0 ?
  10. Line 131 before Equation (13) - Whether it should not be specified in what time period the average work is determined? Independence on the acceleration interval (and whence also dist0) results from the derived formula (13). It is a conclusion, not an assumption.
  11. It is not clear what Equation (39) describes? If it concerns forces fH(t), Eq. (37) and fM(t), Eq. (38), in view of further description of these forces, should have An and Bn coefficients.  If it describes the form of the Fourier series, in general, isn't it better to quote it before the first use of the Fourier series in the text?
  12. Wouldn't it be better to consistently label the functions associated with the "man" and "woman" cases? For Example, notations in Figure 5 and the labels in forces  fH(t) and fM(t) are inconsistent.
  13. Is the use of different terms – kayak (the title, abstract, keywords, etc. ), canoe (abstract, line 115), vessel (lines 133, 134, 156), boat (lines 107, 259) - for the boat justified?
  14. Equation (42) – the second term of the right-hand side. Does the coefficient is b12, not b22 ?
  15. Equations after line 323 - Is the equality for n=0 correct? Where A0/2 ?
  16. Sine function - The notation "sin", not "sen" appropriate for the Spanish-language areas, would be more appropriate here; Equation (17), and the following. In Equations (32)-(34) notation "sin" is used, but soon after - the caption for Figure 6 and Equation (36) - "sen" returns.
  17. The units - various abbreviations are used, also in the figures and in their captions. For example, Figure 7: "seconds" - horizontal axis, "s" - vertical axis, "sec" - caption of the figure. Please be strict with units and dimensions.
  18. Some symbols are not precisely defined; although, based on the text, you can guess what quantities they describe. For example, fave- Equation (31).
  19. There seems to be a lack of necessary precision concerning editorial aspects of preparing the manuscript. The shortcomings that I noticed mainly include:
  • Symbol notations are used inconsistently. For example: for ta, Equation (7), in the caption of Figure 3 notation t99%, while in lines 143, 149 notation t99 are used.
  • Line 287 – Table 3 or Table 1?
  • Caption of Figure 6 - can be probably written more succinctly.
  • Caption of Figure 7 –mass is denoted previously by “m”.
  • Tables 2 and 3 - the reversed order makes reading difficult and is also inconsistent with the guidelines.
  • The lines that follow some equations start with a capital letter and indent, but they shouldn't. For example: after Equations (1)-(3), (5), (7), (12), (32)-(34, (36). I understand this is an oversight, but it should be corrected.

With best regards,

Reviewer

Author Response

We would like to thank the referee for his/her comments. In the attached file we will address the comments and describe the corrections introduced in the manuscript.

Reviewer 2 Report

Dear authors,

I consider the manuscript a thorough study on the topic of mathematical modelling of paddling dynamics. The manuscript makes a convincing case for the novelty and scientific value of the proposed modelling approach. The developed model and simulation results are presented clearly are compared to previous work thoroughly.

My comments concern grammar and formatting:

-No period or colon in headings or subheadings.

-Table caption should be placed above the table.

-When an equation is followed by “where . . . ” the, “where” should be lower case and left aligned because it does not begin a new paragraph. For example in lines 82, 89 and 103.

-Similarly, the phrase “in the case of men, and . . .” after equation 37, does not begin a new sentence or a paragraph and should be left aligned.

-Words in equations should not use italic font. For example, “with” in equation 3.

-The MDPI Style Guide recommends to punctuate equations: “Punctuate equations as part of a regular sentence. For example, if the equation comes at the end of a sentence, a period should be placed immediately after the equation.”

-Some sentences are lengthy and difficult to comprehend. Please consider splitting them to simpler and shorter sentences. For example, the sentences in lines 225-231, 204-209 and 335-344.

-When using references, note that “et al.” is an abbreviation and should have a period after “al”.

-The unit symbol of kilogram “kg” should not be capitalized in the captions of figures 1 and 2 or elsewhere throughout rest of the manuscript.

-Please don’t begin a new sentence with “And”. For example in lines 306 and 311.

-Please consider splitting the section 4 “Estimations of the model” to subsections with descriptive headers. In the current form it lacks a coherent structure.

-Line 107: The vessel is here referred to as “boat” but elsewhere a “kayak”. Is this the correct use of terms?

-Line 132: Should this say “approximate”?

-Line 178: Seems there should be a colon at the end of the sentence instead of a period.

-Line 191: What does “has no relevant paper” mean?

-Line 199: The word “We” should not be capitalized, because it comes after a comma.

-Line 223: Correct spelling is “olympic”.

-Line 235: The phrase ". . .as a result it does the velocity” is confusing.

-Line 316: In this sentence “can neglected” should say “can neglect”.

-Line 362: The number of “Treus et al.” in the reference list is missing.

Author Response

We would like to thank the referee for his/her comments. In the attached file we will address the comments and describe the corrections introduced in the manuscript.  Most of these corrections and the suggested by the other referees are highlighted in blue in the manuscript.

Thanks for the comments, we believe the paper has improved after introducing the reply to your comments.

Reviewer 3 Report

Dear Authors!

I have carefully read your certainly interesting manuscript. I want to note the paper is comprehensive and mathematically elegant.
The article provides a mathematical description of the kayak movement, considering propulsion force constant and periodic. Different models of paddle strokes are presented and verified. One of the most important contributions of the paper is the novel approximation of the stroke using Fourier series which allows obtaining realistic velocity prediction, confirmed by comparison with practical measurements. This allows the researcher to take into account the arbitrary stroke profile and compare the results between different theoretically possible profiles that could be realized in practice by adjusting the rowing technique or using a different blade shape.
Also, the strength of the work is the use of up-to-date data, since a significant part of the articles in references was published in the last 5 years. This is partly due to the fact that it is only in recent years that DAQ tools have become available to perform accurate measurements in sports.

I also would like to point out some of the flaws.
The function 'sen', instead of 'sin', is encountered several times. Please fix.
There are periods after the headings, which should not be done.

These disadvantages do not detract from the merits of the work. I believe that it can be published after minor revision.

Author Response

We would like to thank the referee for his/her comments. In the attached file we will address the comments and describe the corrections introduced in the manuscript.

  regards

Authors

Round 2

Reviewer 1 Report

My comments are in the enclosed pdf file.

Author Response

We thank the reviewer once again for his comments and apologize for the first round's oversights. In the file attached we describe the latest changes to the article. 

In this second round we have marked in red the changes introduced.

Regards,

The authors
